# Intraflagellar transport drives flagellar surface motility

**Sheng Min Shih[1], Benjamin D Engel[2], Fatih Kocabas[3†], Thomas Bilyard[1], Arne Gennerich[4], Wallace F Marshall[2], Ahmet Yildiz[1,5]\***

[1]Department of Physics, University of California, Berkeley, Berkeley, United States; [2]Department of Biochemistry, University of California, San Francisco, San Francisco, United States; [3]Department of Internal Medicine, University of Texas Southwestern Medical Center, Dallas, United States; [4]Anatomy and Structural Biology, Albert Einstein College of Medicine, Bronx, United States; [5]Department of Molecular Cell Biology, University of California, Berkeley, Berkeley, United States

**Abstract** The assembly and maintenance of all cilia and flagella require intraflagellar transport (IFT) along the axoneme. IFT has been implicated in sensory and motile ciliary functions, but the mechanisms of this relationship remain unclear. Here, we used *Chlamydomonas* flagellar surface motility (FSM) as a model to test whether IFT provides force for gliding of cells across solid surfaces. We show that IFT trains are coupled to flagellar membrane glycoproteins (FMGs) in a $Ca^{2+}$-dependent manner. IFT trains transiently pause through surface adhesion of their FMG cargos, and dynein-1b motors pull the cell towards the distal tip of the axoneme. Each train is transported by at least four motors, with only one type of motor active at a time. Our results demonstrate the mechanism of *Chlamydomonas* gliding motility and suggest that IFT plays a major role in adhesion-induced ciliary signaling pathways.

**\*For correspondence:** yildiz@
berkeley.edu

**†Present address:** Texas Institute of Biotechnology, North American College, Houston, United States

## Introduction

Cilia and flagella are microtubule-based organelles that power the locomotion of many organisms, generate fluid flow over multiciliated surfaces, and mediate cell signaling (*Liem et al., 2012*). In order to assemble and maintain cilia, ciliary proteins are transported from cytoplasm to the tip by IFT along axonemes (*Kozminski et al., 1993*). In IFT, linear arrays of multiprotein complexes (IFT trains) are transported by kinesin-2 and dynein-1b in anterograde and retrograde directions, respectively (*Cole et al., 1998*; *Porter et al., 1999*). IFT is a universal mechanism for nearly all eukaryotic cilia and flagella, and defects in this process are linked to a wide range of human diseases, including polycystic kidney disease, retinal degeneration (*Rosenbaum and Witman, 2002*; *Ishikawa and Marshall, 2011*), and Bardet-Biedl syndrome (*Ou et al., 2005*; *Lechtreck et al., 2009*, *2013*; *Wei et al., 2012*).

Several studies have suggested that IFT not only transports material between the cell body and the flagellar tip, but also interacts dynamically with the flagellar membrane (*Kozminski et al., 1993*) to regulate diverse ciliary functions including motility, mating, sensing extracellular signals and influencing developmental decisions (*Huangfu et al., 2003*; *Snell et al., 2004*; *Pedersen and Rosenbaum, 2008*; *Ishikawa and Marshall, 2011*). However, it has remained unclear how transport of IFT trains underneath the flagellar membrane transmits force to components at the exterior of the flagellar membrane.

In order to investigate interactions between IFT and the ciliary surface, we used *Chlamydomonas reinhardtii* gliding motility as a model system. In *Chlamydomonas*, the flagellar surface is highly dynamic; polystyrene microspheres and other inanimate small objects adhere to and are moved bidirectionally

**eLife digest** Cilia and flagella protrude like bristles from the cell surface. They share the same basic '9+2' axoneme structure, being made up of nine microtubule doublets that surround a central pair of singlet microtubules. Flagella are generally involved in cell propulsion, whereas motile cilia help to move fluids over cell surfaces.

Maintaining cilia and flagella is a challenge for cells, which must find a way to send new proteins all the way along the axoneme to the site of assembly at the flagellar tip. Cells achieve this via a process called intraflagellar transport, in which proteins are carried back and forth by kinesin and dynein motors along the axonemal doublet microtubules. Intraflagellar transport has been proposed to influence other functions of cilia and flagella, including the propulsion of cells over surfaces. However, the details of these interactions are unclear.

Through a combination of biophysical and microscopy approaches, Shih et al. describe the mechanism that the green alga *Chalmydomonas* uses to power flagellar gliding over surfaces. By tracking single fluorescently tagged molecules, Shih et al. observed that flagellar membrane glycoproteins are carried along the axoneme by the intraflagellar transport machinery. During transport, flagellar membrane glycoproteins transiently adhere to the surface, and dynein motors that were previously engaged in carrying these glycoproteins now transmit force that moves the axonemal microtubules. This process, which is dependent on the concentration of calcium ions in the extracellular environment, generates the force that propels the alga's flagella along the surface.

Gliding motility is thought to have been one of the initial driving forces for the evolution of cilia and flagella. How the intricate mechanism of flagellar beat motility could have evolved has been the subject of much discussion, as it would require the flagellum to have evolved first. In demonstrating that gliding motility is powered by the same intraflagellar transport mechanism that is required for flagellar assembly, Shih et al. provide strong evidence for the evolution of primitive flagella before the evolution of flagellar beating. Furthermore, since algal flagella have essentially the same structure as the cilia of human cells, these findings could ultimately aid in the development of treatments for diseases that result from defects in intraflagellar transport, including polycystic kidney disease and retinal degeneration.

along the flagellar surface, and cells glide over solid surfaces via adhesion of their flagella (*Lewin, 1952*; *Bloodgood, 1981*). Gliding motility is central to understanding the function and evolution of cilia, as it may have existed in early cilia before the establishment of axonemal beating. There are indications that gliding and FMG1-B motility are driven by the same process (*Bloodgood, 2009*), because they move at comparable speeds and both require ligation of the major flagellar surface protein, FMG1-B, into large clusters (*Bloodgood and Workman, 1984*), accompanied by a $Ca^{2+}$-dependent signaling pathway (*Bloodgood and Salomonsky, 1994*). Microsphere movement is considered a more easily assayed and quantitated surrogate for the force transduction system that drives whole cell gliding motility. However, gliding is driven by the pulling motion of the leading flagellum, whereas FMG1-B movement is bidirectional (*Bloodgood, 2009*), implying that the two motilities could employ different motors.

It has been proposed that IFT provides the force for gliding motility (*Bloodgood, 2009*). IFT trains make multiple connections with the flagellar membrane (*Pigino et al., 2009*) and carry several ciliary and flagellar membrane proteins (*Qin et al., 2005*; *Huang et al., 2007*; *Lechtreck et al., 2009*, *2013*). Inactivation of kinesin-2 in the temperature-sensitive (*ts*) mutant *fla10^ts* stops both IFT and gliding motility (*Kozminski et al., 1995*). While these results suggest that kinesin-2 serves as the anterograde motor responsible for both microsphere movement and gliding motility (*Kozminski et al., 1995*; *Laib et al., 2009*), the retrograde motor for these motilities has not been clearly identified. Mutations in the LC8 subunit of dynein do not abolish FMG1-B movement (*Pazour et al., 1998*), and other flagellar motors, such as the minus-end directed kinesin KCBP (*Dymek et al., 2006*), have been proposed to drive FSM (*Bloodgood, 2009*).

Several studies have raised arguments against this model. IFT motility differs significantly from FSM in that trains move faster and more processively along the length of the flagellum (*Kozminski et al.,*

*1993*; *Bloodgood, 2009*). FSM requires micromolar levels of free calcium, whereas IFT is $Ca^{2+}$-independent (*Kozminski et al., 1993*; *Bloodgood, 2009*). Sexual agglutinins were observed to migrate from the cell body into flagella in the absence of IFT (*Pan and Snell, 2002*). Therefore, evidence supporting the role of IFT in gliding motility is indirect and the precise functions of IFT, molecular motors, FMG1-B and $Ca^{2+}$ in FSM remain unclear.

## Results

To dissect the mechanism of FSM, we directly observed IFT, FMG1-B and gliding motilities using single-molecule imaging techniques. We monitored the movement of individual IFT trains by using total internal reflection fluorescence (TIRF) illumination to image paralyzed-flagella (*pf*) mutant cells that had adhered to the glass surface with both flagella. To establish a link between FMG1-B transport and IFT, we simultaneously tracked the movement of FMG1-B antibody-coated fluorescent beads (200 nm diameter, dark red) and IFT27-GFP in the *pf18* strain. The beads performed short processive runs with reversals of direction whereas IFT trains moved in a regular and unidirectional manner. Multicolor kymography analysis shows that the beads transiently dissociated from one IFT train, diffused for a period of time and then bound to another IFT train (*Figure 1A*, *Video 1*) similar to the movement of extraflagellar particles observed along the flagellar membrane (*Dentler, 2005*). Diffraction-limited images of beads and IFT trains were fitted to a two-dimensional Gaussian to achieve a higher localization precision (*Yildiz et al., 2003*). *Figure 1B* shows the colocalization of a membrane-attached bead with an individual IFT train as it moves unidirectionally for >500 nm. The movements of the bead and the colocalized IFT trains correlate strongly (>0.99) with each other during the processive run (N = 30, see *Figure 1—figure supplement 1* for more examples), excluding the possibility that microspheres coincidentally moved together with IFT trains. We measured the distance between microspheres and the nearest IFT trains moving in the same direction for both the experimental data and randomly generated kymographs. The Kolmogorov-Smirnov test (p=7 × 10$^{-9}$, *Figure 1C*) demonstrates that IFT trains colocalize with FMG1-B along the flagellar membrane (*Figure 1D*).

We next investigated whether IFT plays a direct role in gliding motility by monitoring individual IFT trains in IFT20-GFP and IFT27-GFP *pf18* cells as they glided on coverslips. A small fraction of IFT trains displayed rare pauses (0.125 s$^{-1}$ per cell, $N_{cells}$ = 23, $N_{pauses}$ = 247) as they moved either in the retrograde or anterograde direction. Remarkably, the pausing of IFT trains was required for the initiation of whole-cell movements (*Figure 2A*, *Figure 2—figure supplements 1 and 2*). We did not observe the start of a gliding event without a paused IFT train. We were able to determine the directionality of 60% of the paused IFT trains, and all of the trains correspondent to the initiation of gliding motility stopped moving during retrograde transport (N = 148). To rule out the possibility that IFT pausing events and initiation of gliding motility are simply coincidental, we quantified the lag time between the pausing of the last retrograde IFT train and the initiation of gliding motility. The average lag time was 0.48 ± 0.05 s. In comparison, we observed individual IFT pausing events (including both anterograde and retrograde pauses) every 8.25 ± 0.96 s per cell (*Figure 2—figure supplement 3*). The Student's *t*-test excludes the null hypothesis that the timing of retrograde IFT pausing and gliding motility are independent of each other (p<0.0001). Anterograde trains also displayed rare pauses (~0.02 s$^{-1}$ per cell, N = 27), but we never observed pausing of an anterograde train before the initiation of gliding motility.

In 24% of all cases (N = 50), a single paused train appeared sufficient to pull the entire cell (*Figure 2A*, *Video 2*). In other cases, multiple trains paused before the start of gliding motility (*Figure 2B*, *Video 2*). Pauses occurred in the leading flagellum, and cells moved in the opposite direction relative to the transport trajectory of the paused trains. In uniflagellate IFT27-GFP *pf18* cells, we also always observed pausing of single or multiple IFT trains before the initiation of gliding motility (*Figure 2—figure supplement 4*). Gliding motility either stopped when paused IFT trains resumed movement (*Figure 2A*), or continued until the cell bodies reached the paused trains (*Figure 2B*). We never observed the cell body gliding further than the position of the paused IFT train(s).

We found compelling evidence that dynein-1b is the primary motor responsible for gliding motility. First, uniflagellate *pf18* cells always glided with the flagellum leading the cell body at 1.49 ± 0.10 µm/s (mean ± SEM) (*Figure 2C–E*, *Video 3*) (*Bloodgood, 2009*), suggesting that gliding motility is driven by pulling forces generated through a minus end directed microtubule motor. In contrast to *pf18* cells, uniflagellate dynein-1b *ts* cells (*dhc1b-3$^{ts}$*) (*Engel et al., 2012*) were unable to display robust gliding at restrictive temperatures (*Figure 2C*, *Video 3*). We observed only short-range (<500 nm) gliding-like

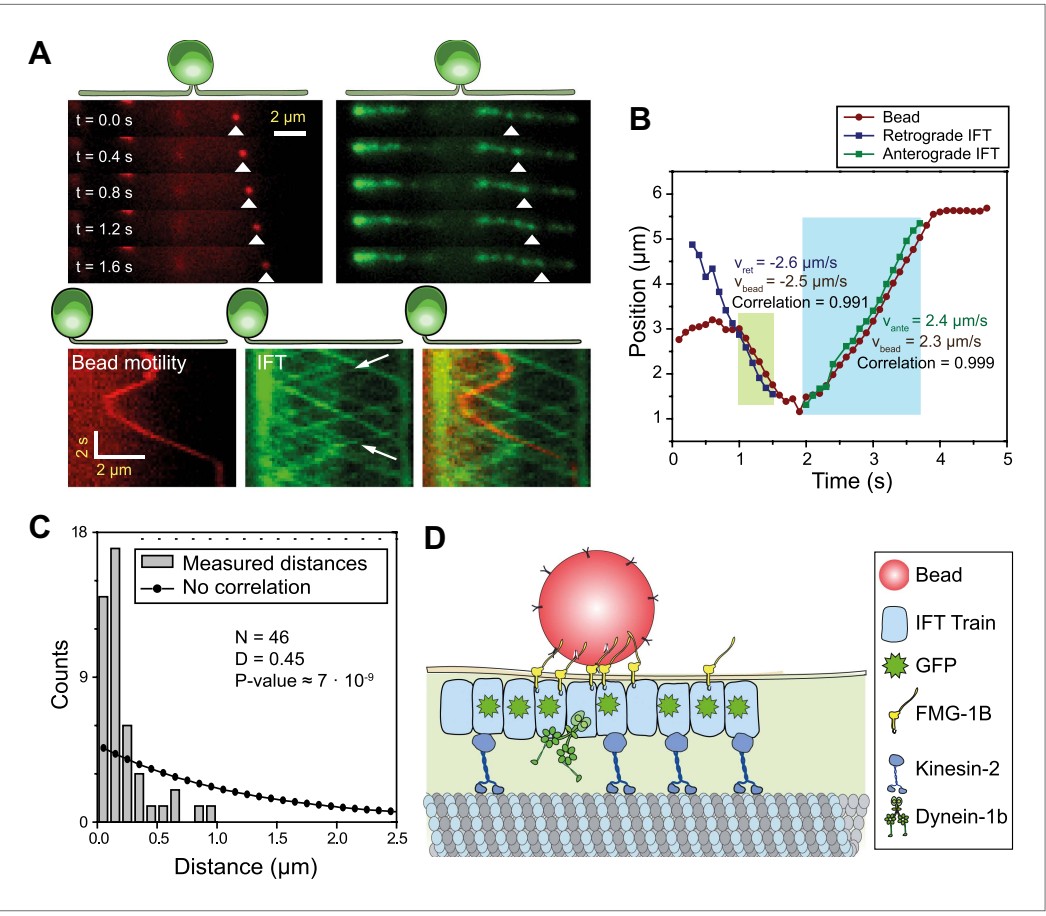

**Figure 1**. IFT transports FMG1-B. (**A**) (Top) Simultaneous tracking of anti-FMG1-B beads (red) and IFT27-GFP (green). (Bottom) Kymographs show that bead motility colocalizes with IFT trains during processive runs. Between the runs, the bead transiently attaches to and detaches from IFT trains. The white arrows indicate the IFT trains transporting the bead. (**B**) Two-dimensional Gaussian fitting of the bead and IFT trains show that bead motility correlates strongly (>0.99) with the movement of individual anterograde (green shaded region) and retrograde (blue shaded region) IFT trains. The bead moves at similar speeds to IFT trains in both anterograde and retrograde directions. (**C**) Comparison of distances from beads to the closest IFT train moving in the same direction (grey bars) to the predicted distribution without correlation (null hypothesis, black line). Kolmogorov-Smirnov statistics indicate that bead and IFT train positions strongly correlate with each other. (**D**) A model for IFT particles transporting FMG1-B. The bead is attached to FMG1-B in the flagellar membrane through antibody linkages (not to scale).

The following figure supplements are available for figure 1:

**Figure supplement 1**. Additional examples of simultaneous tracking of bead motility and IFT.

motion in 54% of these cells, compared to robust unidirectional gliding motility observed over 5 μm in *pf18* cells. These short-range motions were bidirectional (*Figure 2D*) and significantly slower (forward: 0.35 ± 0.09 μm/s, backward: 0.47 ± 0.16 μm/s) than the gliding speed of *pf18* cells (*Figure 2E*). Second, inactivation of dynein-1b in biflagellate *dhc1b-3*[ts] cells decreased the fraction of gliding cells from 100% to 52% and reduced the gliding speed from 0.86 ± 0.08 μm/s to 0.18 ± 0.03 μm/s. In contrast, inactivation of kinesin-2 in *fla10*[ts] cells resulted in a twofold increase in gliding speed (*Figure 2E*), indicating that kinesin plays an inhibitory role during dynein-1b driven gliding motility. These results agree with our observations that the pausing of anterograde trains does not initiate gliding motility.

To further test the role of dynein-1b in gliding motility, we treated the cells with a small molecule inhibitor of dynein, ciliobrevin D (*Firestone et al., 2012*). We varied ciliobrevin D concentration between 0–150 μM and monitored IFT in cells adhered both of their flagella to surface 2 min after

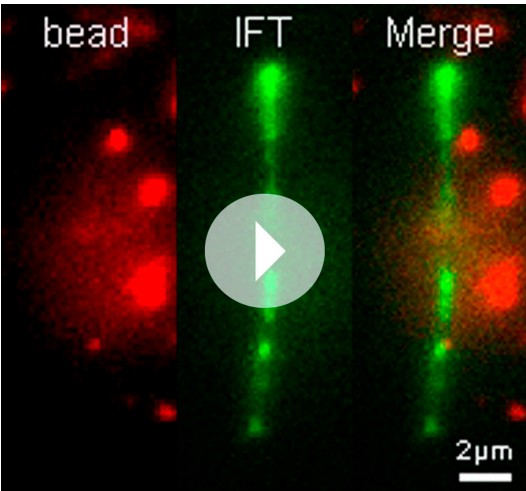

**Video 1**. Simultaneous imaging of bead motility and IFT. The left channel shows fluorescent beads coated with anti-FMG1-B. One bead displays bidirectional movement along the flagellum. The middle channel shows the movement of IFT trains within the IFT27-GFP *pf18* cell. The right channel is the superimposed image of the two channels (IFT: green, bead: red). Movement of the bead colocalizes with that of individual IFT trains. The data was collected at 5 frames/s. The size of a single channel is 6.8 × 19.1 µm.

drug treatment (*Figure 3A*, *Figure 3—figure supplement 1*, *Video 4*). Both anterograde and retrograde IFT train frequencies dropped with increasing concentrations of ciliobrevin D (*Figure 3B*), and at >100 µM ciliobrevin D we observed accumulation of IFT trains at the flagellar tip (see example kymograph in *Figure 3A*). At 150 µM ciliobrevin D, retrograde IFT frequency was reduced by 92% compared to the 70% decrease observed after 6 hr of heat inactivation of *dhc1b-3*[ts] cells (*Engel et al., 2012*). The velocities of retrograde and anterograde trains also decreased by 60% and 36%, respectively (*Figure 3C*). Importantly, inhibition of dynein-1b resulted in a significant reduction in the speed (79%) and frequency (79%) of gliding motility (N = 50, *Figure 3D*). These results further support our conclusion that dynein-1b motors produce the force for gliding motility.

Based on these results, we propose a model to describe the functions of IFT motors, IFT trains and FMG1-B transport in gliding motility (*Figure 2F*). Surface adhesion of the FMG1-B cargo through its large extracellular carbohydrate domain (*Bloodgood, 2009*) stops the retrograde IFT train. Dynein motors previously engaged in transporting the paused IFT train exert force towards the microtubule minus end, causing the whole cell to move toward the plus-end flagellar tip. Thus, gliding motility works similarly to microtubule gliding assays, in which surface-immobilized dyneins glide microtubules with their plus-end tips in the lead.

To investigate how cells reverse gliding direction, we developed a kymography method for monitoring IFT trains as the cells reorient their flagella during gliding (*Figure 4A–B*, *Video 5*). During 60% of reversals (N = 40), the cell lifted one of its flagella and the paused IFT trains on the surface-adhered flagellum drove the motility (*Bloodgood, 2009*). In other cases, single or multiple paused IFT trains accumulated in one flagellum, and the cell body glided toward this cluster until the paused trains either detached from the glass surface or reached the flagellar base. We also observed cases with paused IFT trains in both flagella where the cell remained immotile, likely due to the balance of forces between bound dynein-1b motors (*Figure 4C–D*). There were no indications of coordination of the pausing events between the two flagella. Different modes of reversals in gliding motility may allow cells to search through the environment by a random walk when they adhere both of their flagella, and to move in unidirectional manner by lifting one of the flagella.

Gliding motility in *C. reinhardtii* is controlled by a $Ca^{2+}$-calmodulin regulated kinase and requires micromolar levels of $Ca^{2+}$ in the media (*Bloodgood and Spano, 2002*). We tested whether the pausing of IFT trains depends on $Ca^{2+}$ (*Figure 5A–B*) by analyzing the kymographs of IFT at different $Ca^{2+}$ concentrations. To quantify pausing in flagella, we used Fourier space direction analysis (FSDA, see *Figure 2—figure supplement 2*) to separate the traces of paused IFT trains and moving IFT trains into two kymographs. Kymographs of paused IFT trains in *Figure 5A,B* show that pausing along the length of the flagellum was significantly reduced in $Ca^{2+}$-deprived cells (see additional examples in *Figure 5—figure supplement 1*). After $Ca^{2+}$ depletion, IFT trains rarely paused in the middle regions of flagella and accumulated at the base (*Figure 5B*, middle). *Figure 5C* plots the total fluorescence intensity of paused IFT trains ($N_{cells}$ = 30 for each case) along the lengths of flagella at different $Ca^{2+}$ levels. The frequency of pausing in $Ca^{2+}$-deprived cells was significantly lower than IFT pausing in cells at a normal $Ca^{2+}$ concentration. In regular TAP media (free $[Ca^{2+}]$ = 0.34 mM), 95% of all surface-adhered cells displayed gliding motility and individual pausing events were observed every ~8 s per flagellum, on average. In contrast, both the fraction of gliding cells and IFT pausing frequency were reduced by

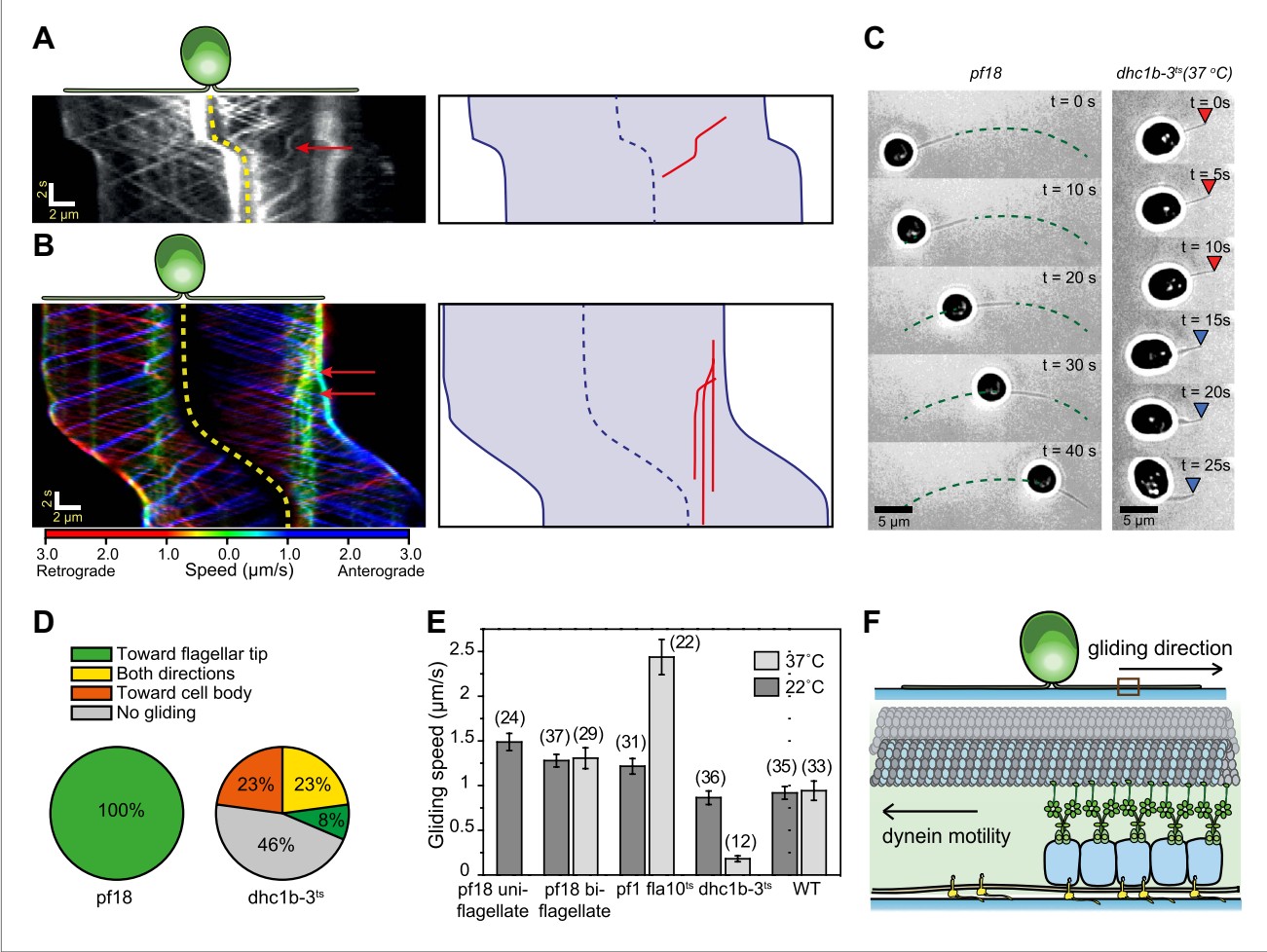

**Figure 2**. Dynein-1b drives gliding motility. (**A**) (Left) Kymograph of a gliding IFT20-GFP cell. A single retrograde IFT train transiently pauses (red arrow) and initiates the gliding movement of the cell toward the paused train. (Right) A schematic representing the timing and trajectory of the paused IFT train (red curve) in the gliding cell shown on the left. (**B**) (Left) Kymograph of an IFT27-GFP cell, pseudo-colored to show the corresponding velocity of each IFT train. Multiple IFT trains (red arrows) pause (green color) prior to gliding motility. The cell glides until it reaches the paused IFT trains. (Right) A schematic representing the timing and trajectories of the paused IFT trains (red curves) in the gliding cell shown on the left. (**C**) Gliding of uniflagellate cells under bright field illumination. A uniflagellate *pf18* cell glides unidirectionally toward its flagellum. A uniflagellate *dhc1b-3ts* cell displays bidirectional gliding at the restrictive temperature. Red and blue arrowheads represent forward (flagellum in the lead) and backward (cell body in the lead) gliding directions. (**D**) All of the uniflagellate *pf18* cells glided with the flagellum in the lead. Heat inactivation of dynein-1b in *dhc1b-3ts* cells at 37°C resulted in a 46% reduction in gliding frequency (N = 35) and led to bidirectional gliding motility. 8% of the cells glided with the flagellum leading the cell body (0.34 ± 0.09 µm/s, mean ± SEM), while 23% glided with the cell body leading the flagellum (0.47 ± 0.16 µm/s). 23% of the cells displayed saltatory gliding movement. (**E**) Uniflagellate cells glided ~20% faster than biflagellate cells. Inhibition of dynein-1b (*dhc1b-3ts*) resulted in a fivefold decrease in gliding speed, whereas inhibition of kinesin-2 (*pf1 fla10-1ts*) led to a twofold speed increase (mean ± SEM). In temperature-insensitive paralyzed (*pf18*) and WT cells, changes in gliding speed between permissive and restrictive temperatures were negligible. (**F**) A model for gliding motility. Retrograde IFT trains adhere to the glass surface through FMG1-B, and the surface-tethered dynein motors pull the cell body through microtubules toward the flagellar tip.

The following figure supplements are available for figure 2:

**Figure supplement 1**. Three additional examples for kymographs of gliding IFT27-GFP cells.

**Figure supplement 2**. The procedure for Fourier space direction analysis.

**Figure supplement 3**. Lag time between IFT pausing and initiation of gliding motility.

**Figure supplement 4**. Three examples for kymograph of gliding uniflagellate IFT27-GFP cells.

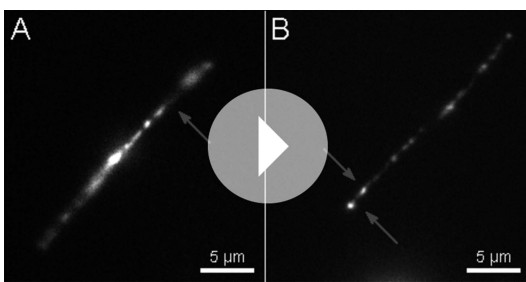

**Video 2**. Pausing of IFT trains during gliding motility. **A**, Gliding motility of an IFT20-GFP cell on a glass surface. A single IFT train tethered to the surface (arrow) immediately preceding the initiation of gliding motility. The size of the window is 24.1 × 24.9 μm. The data was collected at 5 frames/s. **B**, Gliding motility of an IFT27-GFP cell on a glass surface. Multiple IFT trains became surface-tethered (arrows), providing force for gliding motility. The size of the window is 24.5 × 24.9 μm. The data was collected at 5 frames/s.

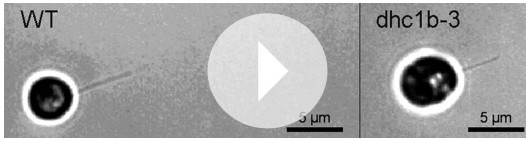

**Video 3**. Gliding motility of uniflagellate WT and dhc1b-3 cells. Uniflagellate WT and *dhc1b-3 Chlamydomonas* cells glide with the flagellum in the lead. The size of the whole window is 47.0 × 11.9 μm. The data was collected at 10 frames/s under a bright-field microscope.

~10-fold at <1 μM $Ca^{2+}$ (**Figure 5D**). Next, we compared the IFT pausing frequencies in gliding and non-gliding cells at different $Ca^{2+}$ levels. The pausing frequency in non-gliding cells was low and independent of $Ca^{2+}$ concentration. In contrast, the pausing frequency in gliding cells was high at normal $Ca^{2+}$ levels and gradually declined to match that of non-gliding cells as the $Ca^{2+}$ concentration decreased below 1 nM (**Figure 5E**).

In $Ca^{2+}$-deprived cells, the beads freely diffused on the flagellar membrane but did not move processively by IFT trains (not shown). Thus, we propose that $Ca^{2+}$ is required for attachment of FMG1-B to IFT trains, not for the activation of a motor protein that provides force for gliding, as previously suggested (**Bloodgood, 2009**). As the flagellar membrane is enriched with a PKD2-like $Ca^{2+}$ channel (**Pazour et al., 2005**; **Huang et al., 2007**), $Ca^{2+}$ signaling at flagellar adhesion sites may play a role in controlling the directionality and timing of gliding movement.

To measure the forces exerted by motors bound to individual IFT trains, we tracked the movement of FMG1-B-antibody coated bead motility using an optical trap. To rule out the possibility that loads exerted by the trap might disrupt the linkage between IFT and FMG1-B, we performed TIRF imaging of IFT27-GFP and optical trapping of beads simultaneously (**Figure 6A**). We observed that IFT trains remained colocalized with trapped beads when subjected to external forces (**Figure 6B**; see **Figure 6—figure supplement 1** for additional examples). The average offset between the positions of the trapped beads and colocalized IFT trains was 280 ± 10 nm (N = 11). This is due to the fact that the beads (920 nm diameter) and IFT trains (200–1000 nm in length) (**Pigino et al., 2009**) are large objects relative to the resolution of conventional fluorescence imaging (approximately 200–250 nm) and the forces that stretch the bead-FMG1-B-IFT linkage displace the center of the bead from the IFT train (see the schematics in **Figure 6—figure supplement 2**). The trap measurements correspond to forces generated by IFT motors and provide direct evidence that IFT transports FMG1-B. Since beads are outside the cell, but are physically linked to the action of IFT trains, the assay combines the advantages of precise in vitro trapping with the ability to manipulate IFT motility under load.

In a fixed trap assay, ~50% of processive bead movements terminated with a stall before returning to the trap center (**Figure 6C**, **Figure 6—figure supplement 3**). We did not observe a significant difference between the force values of stall and release events. This suggests that motors attached to an IFT train may not be able to reach their maximum stall force, defined as the stall force of a single motor multiplied by the number of bound motors (**Shubeita et al., 2008**). Histograms of peak forces (**Figure 6D**) show that anterograde and retrograde IFT trains moved against 21.4 ± 0.7 pN and 25.2 ± 1.3 pN (SEM), respectively. Forces exerted on IFT trains well exceed the force-generation capability of a single motor. Previous studies demonstrated that multiple motors produce larger forces, move with higher velocities under load and carry cargos further than a single kinesin or dynein motor (**Mallik et al., 2005**; **Vershinin et al., 2007**). It remains controversial whether motor stall forces are additive at low motor copy numbers (**Vershinin et al., 2007**) or whether multiple motors tend to transport their cargo using only one load-bearing motor at a time (**Jamison et al., 2010**). Therefore, we believe that peak forces in our experiment represent a lower boundary for estimates of motor copy number. Assuming that IFT motors

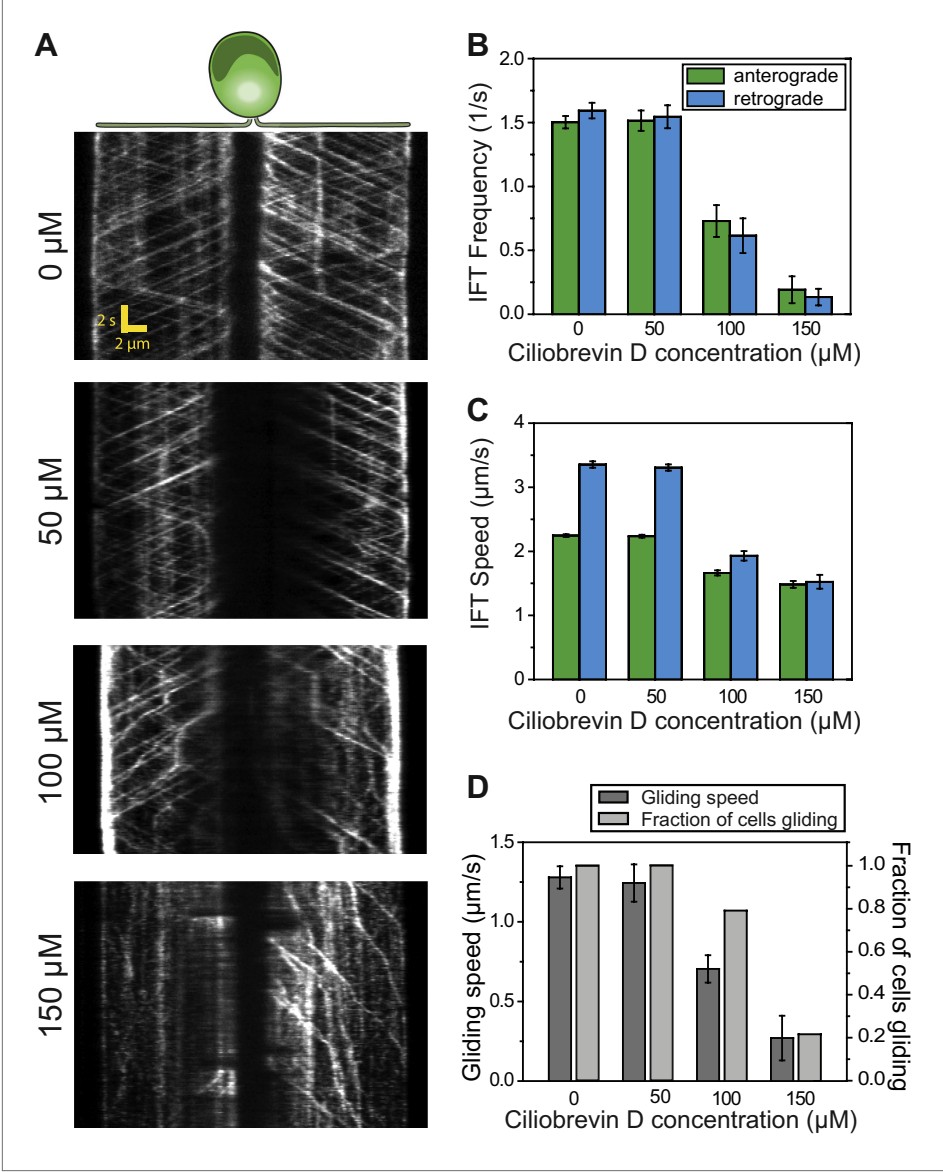

**Figure 3**. Ciliobrevin D inhibits dynein-1b and stops gliding motility. (**A**) Kymographs of IFT20-GFP cells treated with varying concentrations of ciliobrevin D. Images were acquired 5 min after addition of ciliobrevin D. (**B** and **C**) Frequency and speed of retrograde and anterograde IFT trains at different ciliobrevin D concentrations. The frequency of retrograde IFT was reduced by 92% at 150 µM ciliobrevin D. (**D**) The speed and fraction of gliding cells decreased by 79% at 150 µM ciliobrevin D.

The following figure supplements are available for figure 3:

**Figure supplement 1**. Additional example for kymographs of IFT20-GFP cells treated with varying concentrations of ciliobrevin D.

produce 6–7 pN forces (*Gennerich et al., 2007*; *Brunnbauer et al., 2010*), we estimate that at least four motors transport IFT trains at a time, in agreement with previously reported values (*Engel et al., 2009b*). It is possible that multiple motor engagements enhance the run length of individual IFT trains, allowing them to traverse the length of a flagellum. In addition, the fast transport of IFT trains in a viscous cellular environment and IFT's role in FSM may require forces larger than single motor stall forces.

The measured forces in both directions only marginally changed (<20%) under different bead anti-body-coating conditions (*Figure 6—figure supplement 4*). This result argues against the possibility

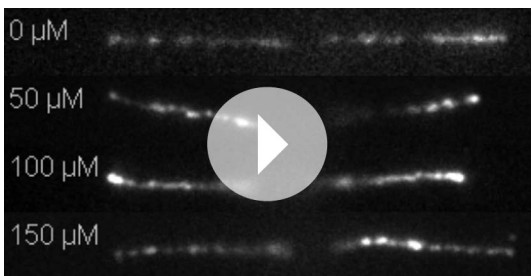

**Video 4**. IFT motility at different concentrations of ciliobrevin D. IFT27-GFP *pf18* cells were immobilized on a glass coverslip. 0–150 µM ciliobrevin D was added to the cell culture and the movies were recorded at 10 frames/s within 2–10 min of drug treatment.

that antibody-coating of the beads leads to the crosslinking of more than one IFT trains, which would be expected to multiply the average peak force. 20% of retrograde runs escaped the trap by producing more than 80 pN of force, presumably via clustering of FMG1-B (*Bloodgood, 2009*), while <1% escape events were observed in the anterograde direction.

We next investigated why the speed of gliding motility (1.49 ± 0.10 µm/s, SEM) is significantly slower than that of retrograde IFT (2.96 ± 0.14 µm/s), even though both processes are powered by dynein-1b motors actively transporting IFT trains. IFT trains that carry 1-µm beads moved ~30% slower than IFT trains that were not associated with the beads (two-tailed Student's *t*-test, p = 2.1 × 10$^{-9}$ (anterograde) and 1.8 × 10$^{-11}$ (retrograde), *Figure 7A*). We reasoned that the viscous drag of the membrane may slow down IFT trains transporting FMG1-B clusters and led to the observed differences in speed between IFT and FSM. To estimate the drag constant of the bead-IFT complex, we analyzed individual stalling events in the optical trap assay (*Figure 7B*) and measured the recoil time of the bead after a stall (*Figure 7C*). The average drag constant of the bead-IFT complex was found to be 8.0 ± 2.8 pN s/µm (SEM, N = 30).

We hypothesized that the large drag constants we measured were due to the interaction between the bead and the membrane. Our results rule out the possibility that microtubule motors step backwards during the recoiling of the bead, as the average bead velocity was 20 µm/s (an order of magnitude faster than that of dynein and kinesin) and there were no detectable backward steps. To rule out other possible interactions between IFT trains and axonemes, we oscillated individual beads on a flagellar membrane surface in a square wave pattern and measured the recoiling time when the beads are decoupled from IFT (*Figure 7—figure supplement 1*). The average viscous drag constant was 8.4 ± 4.2 pN.s/µm, which is similar to the drag that beads experience when they interact with IFT trains. Anterograde and retrograde trains moving at 2–3 µm/s would experience 16–24 pN resistive forces, comparable to the total motor force exerted on a single IFT train (*Figure 7D*). Therefore, IFT is subjected to a high drag force when it carries a large bead along the flagellar surface, which leads to the reduction of transport velocity.

We next investigated how kinesin-2 activity reduces the speed of gliding motility (*Figure 2E*). This could theoretically be explained by a tug-of-war between active kinesin and dynein motors that are both present on the same IFT train. Alternatively, kinesin-2 and dynein-1b motors may be exclusively active on anterograde and retrograde cargos, respectively, and pausing of anterograde trains could produce forces that oppose the gliding forces of paused retrograde trains. To distinguish between tug-of-war and coordinated transport mechanisms, we examined how inhibition of one class of motors affected the forces exerted on IFT trains traveling in the opposite direction (*Laib et al., 2009*). At permissive temperatures, the peak forces of IFT in *fla10-1^ts^* and *dhc1b-3^ts^* were in close agreement with wild-type (WT) cells. Heat-inactivation of kinesin-2 reduced the frequency of anterograde runs by 66%, but did not alter the peak forces on retrograde runs (*Laib et al., 2009*) (*Figure 8A*). Similarly, heat inactivation of dynein-1b significantly reduced the ratio of retrograde to anterograde transport events (0.15), but had minimal effect on the peak forces of anterograde runs (*Figure 8B*). These results exclude a tug-of-war mechanism in IFT, which would lead to an increase of force from motors walking in one direction upon inactivation of the motors walking in the opposite direction. Instead, only one type of a motor remains active on IFT trains at a time, which is consistent with the result that retrograde speed of *fla10-1^ts^* does not change after kinesin-2 is inactivated. We propose that kinesin-2 on paused anterograde trains slows down gliding motility by exerting forces in the opposite direction to that of dynein-1b on paused retrograde trains.

## Discussion

We present direct evidence for the mechanism of IFT-mediated cell-surface interactions using *Chlamydomonas* gliding motility and FMG1-B transport as a model system. In *Chlamydomonas*, IFT

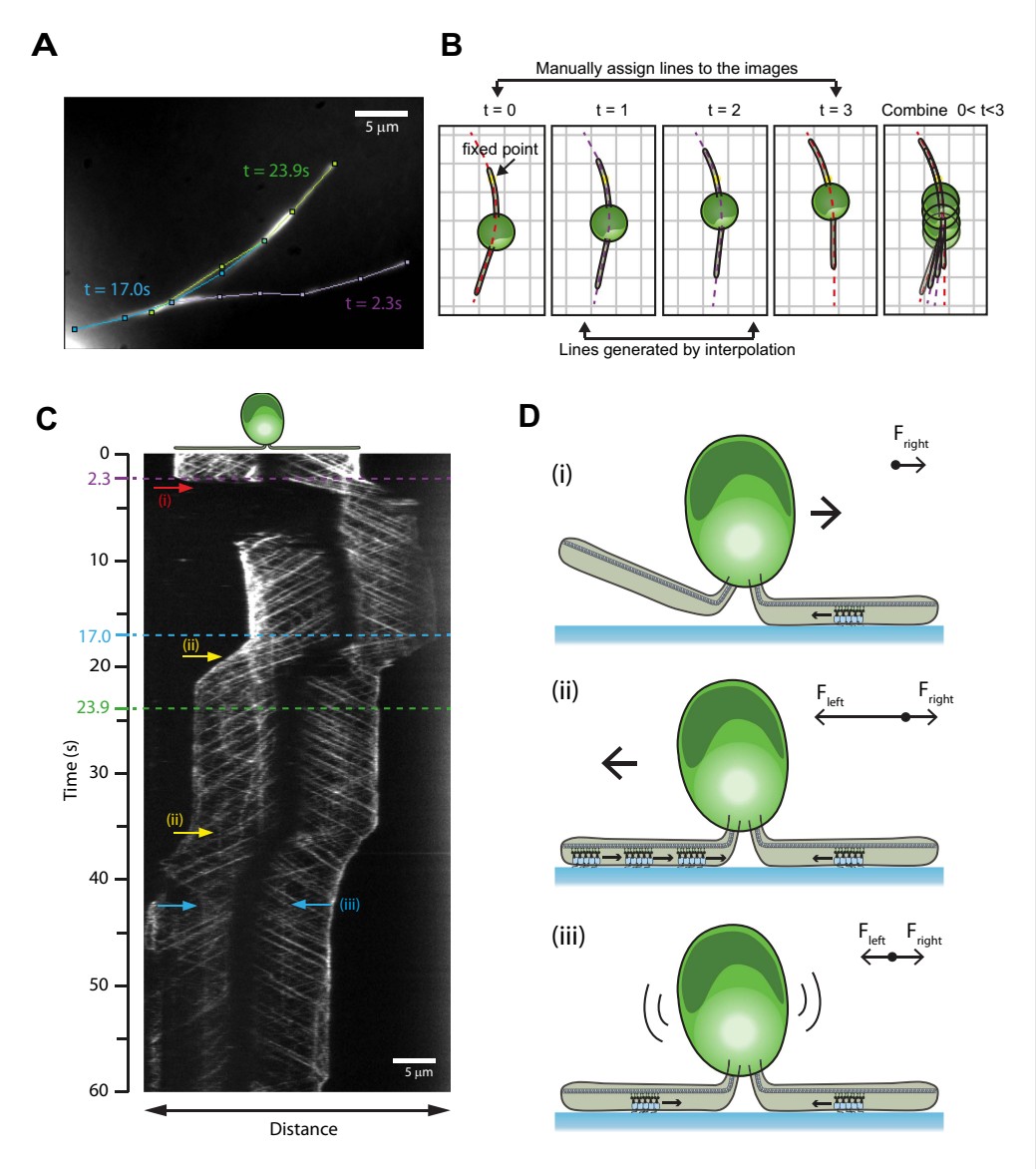

**Figure 4**. Mechanisms of reversal in gliding motility. (**A**) The average intensity of all the frames in the first 60 s of **Video 5** of an IFT27-GFP *pf18* cell, showing the path of the gliding flagella. (**B**) To monitor IFT trains while the cells alter their flagellar orientation in gliding motility, different curves were plotted before and after cells reoriented their flagella (red dotted lines). Each curve shares at least one common point with previous and subsequent curves. Intermediary frames were analyzed by interpolation of the assigned curves, and the intensity along the line in each frame was stacked according to the fixed point (yellow dot). (**C**) A kymograph generated by this method reveals multiple ways that the cell can control gliding direction. (i) When the cell lifts one flagellum (red arrow), the cell body moves toward the surface-attached flagellum. When both flagella are attached to the surface, gliding direction is determined by the balance of forces exerted by surface-tethered IFT trains. (ii) The cell glides toward the flagellum with more paused IFT particles (yellow arrow). (iii) When there are equal numbers of surface-tethered IFT particles in both flagella (cyan arrow), opposite forces cancel out and the cell remains stationary. (**D**) A schematic model representing the three different modes (i, ii, iii) of gliding direction reversal observed in (**C**).

trains carry FMG1-B as a cargo. The interaction between IFT trains and FMG1-B clusters is transient, as FMG1-B boards moving trains but usually unloads and diffuses away before the trains arrive at the flagellar base or tip. During gliding motility, IFT trains pause due to attachment of the FMG1-B cargo to the surface, and the dynein-1b motors engaged to the paused IFT trains generate pulling forces

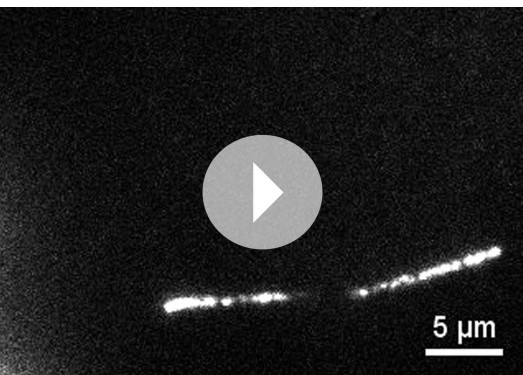

**Video 5**. Reversal of direction in gliding motility. Gliding motility of an IFT27-GFP cell on a glass surface. The cell changes the direction and speed of gliding motility either by raising one of its flagella or by the pausing of one or more IFT trains relative to the glass surface. The data was collected at 10 frames/s. The size of the window is 33.9 × 23.7 µm.

along the microtubule long axis. While the majority of organismal motilities rely on the actin cytoskeleton or axonemal beating, gliding motility in *C. reinhardtii* is distinct in that it is powered by intracellular transport machinery along microtubules. By providing surface adhesion points, FMG1-B performs an analogous function to integrins in mammalian cell motility (*Bloodgood, 2009*; *Lecuit et al., 2011*).

Our results also demonstrate how different types of flagellar motility in *Chlamydomonas* have distinct characteristics despite being driven by the same forces. First, we showed that viscous drag of the membrane slows down the motility of IFT trains that carry FMG1-B-conjugated beads, causing gliding motility and FMG1-B transport to proceed slower than IFT. Second, previous work showed that IFT is unaffected by changes in environmental $Ca^{2+}$ concentration (*Bloodgood and Salomonsky, 1990*; *Kozminski et al., 1993*), whereas gliding motility is $Ca^{2+}$-dependent (*Bloodgood and Salomonsky, 1990*; *Kozminski et al., 1993*). This raised the possibility that $Ca^{2+}$ may be required to activate a specific motor protein that provides force for gliding (*Bloodgood, 2009*). Our results are inconsistent with this hypothesis. We found that the presence of free $Ca^{2+}$ leads to frequent pauses in IFT motility at flagellar adhesion sites. In the absence of $Ca^{2+}$, IFT moves unidirectionally without interruptions, suggesting that attachment of FMG1-B to IFT trains is regulated by a $Ca^{2+}$-dependent signaling pathway. The signaling pathway responsible for linking FMG1-B to IFT remains unclear. There is evidence that surface adhesion or crosslinking of FMG1-B into large clusters induces dephosphorylation of a transmembrane protein that co-immunoprecipitates with FMG1-B (*Bloodgood and Salomonsky, 1994*, *1998*). The complex may be regulated by a calcium-calmodulin dependent gliding associated kinase (GAK), which is required for the gliding motility (*Bloodgood and Spano, 2002*). Further investigation is required to identify the rest of the signaling pathway regulating IFT-flagellar surface interactions.

Our force measurements show a number of similarities and differences with earlier published work. In a previous trapping assay, beads nonspecifically adsorbed to the flagellar membrane were found to exert 60 pN average peak forces (*Laib et al., 2009*). In contrast, our measured peaked forces are significantly lower (20–30 pN) than these values. Laib et al. could not determine whether their force measurements reflect the forces produced by motors attached to IFT trains or by motors that are directly bound to FMG1-B clusters. Our experiments directly link optical trap recordings of bead motility to forces generated by flagellar motors attached to IFT trains (*Figure 1*), enabling us to propose physical models for the coordination of IFT and its role in FSM. In agreement with Laib et al., we observe reciprocal coordination of motors during anterograde transport. Additionally, our measurements of dynein-1b inactivation verify that motors do not engage in a tug-of-war while travelling in the retrograde direction.

The mechanism we described for IFT pausing and force generation has broad implications for the traffic of ciliary sensory proteins as well as cell signaling at ciliary membrane adhesion points (*Wang et al., 2006*). For example, IFT is required for signal transduction during the mating of ciliated organisms. The initial interaction between gametes of two mating types in *Chlamydomonas* can occur at any point along their flagellar membranes, but the tips of their flagella must be aligned and locked before activating gametic fusion (*Homan et al., 1987*). Since IFT trains transport sexual agglutinins within the flagellar membrane of gametic cells (*Ferris et al., 2005*), adhesion between sexual agglutinins of opposite cell types may stall retrograde IFT movement. As a result, forces produced by IFT trains would move the flagellar contact points until a balance of forces is achieved when the flagella are properly aligned with respect to each other. Indeed, microspheres have been observed to move and accumulate at the flagellar tips during mating (*Hoffman and Goodenough, 1980*). We propose that the absence of retrograde bead movement may be related to the formation of adhesion contacts between the flagella and pausing of retrograde IFT trains.

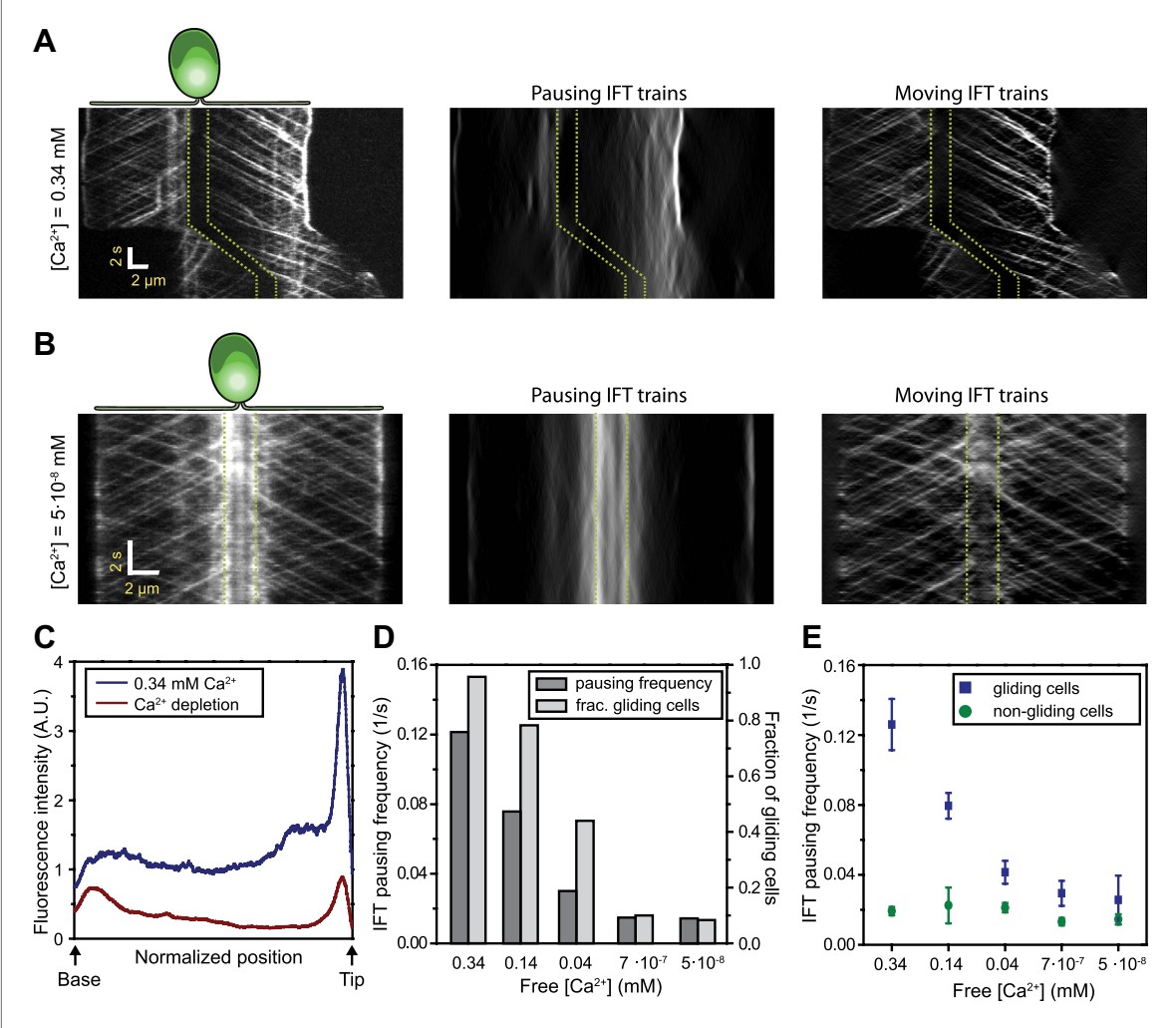

Figure 5. Ca²⁺ is required for the pausing of IFT trains at flagellar adhesion sites. (**A**) (Left) Kymograph of an IFT27-GFP cell adhering both of its flagella in the presence of 0.34 mM free Ca²⁺. Pausing (middle) and moving (right) IFT trains were split into separate kymographs by FSDA analysis. IFT trains pause frequently along the length of the flagellum and drive gliding motility. (**B**) When cells are deprived of free Ca²⁺, immotile IFT trains accumulate near the cell body and do not display frequent pauses between the flagellar base and the tip. (**C**) The average fluorescence intensity of 'pausing IFT trains' in kymographs relative to the length of the flagellum (N = 6 cells for each condition). Cells in 0.34 mM Ca²⁺ show robust pausing uniformly along the length of the flagellum. In contrast, Ca²⁺-deprived cells show significantly reduced pausing events. Background fluorescence was excluded from the analysis. (**D**) IFT pausing frequency and the fraction of gliding cells as a function of free Ca²⁺ concentration. The free Ca²⁺ concentration was controlled through Ca²⁺-EGTA buffering. (**E**) IFT pausing frequency of gliding and non-gliding cells as a function of free Ca²⁺. At 0.34 mM Ca²⁺, the IFT pausing frequency in gliding cells is very high compared to non-gliding cells. In Ca²⁺-deprived cells, the pausing frequency in gliding cells is reduced to the residual level of pausing events observed in non-gliding cells.

The following figure supplements are available for figure 5:

**Figure supplement 1**. The analysis of IFT pausing in the presence and absence of free Ca2+ in media.

Cilia in the retina, liver, and kidney cells were recently observed to make direct physical contacts, which may serve as 'bridges' for signaling networks between many cells (***Ott et al., 2012***). These contacts are tight adhesions between the ciliary membranes, mediated by N-linked glycoproteins. It is possible that IFT exerts force on cell-cell adhesion sites and determines the positioning of these adhesion sites by moving them along the length of the cilium. Ca²⁺ signaling at flagellar adhesion sites may play a major role in regulating attachment to IFT trains and controlling the direction of force generation. We believe that the assay we developed will be a starting point for deciphering the role of IFT in the signaling, mating and development of ciliated cells.

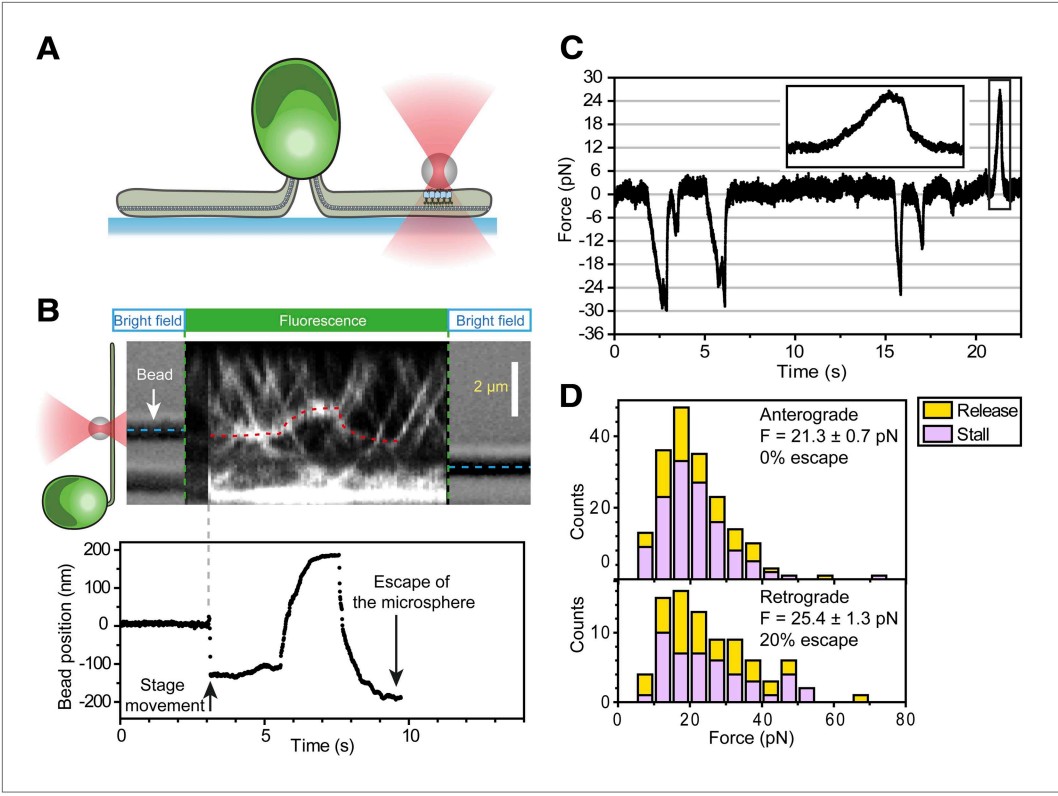

**Figure 6**. Stall force measurements on single IFT trains. (**A**) Schematic representation of combined optical trapping of bead motility and fluorescent tracking of IFT. (**B**) Simultaneous tracking of IFT27-GFP and bead motility. At t = 3 s, the microscope stage was moved to bring the flagellum underneath the trapped bead. An IFT train stalls at the trap position (t = 5–8 s) and stays adhered to the bead until the bead escapes the trap (t = 10 s). The CCD camera is toggled between fluorescence and bright field imaging to monitor IFT trains as well as the bead when it is out of the detection range. (**C**) Displacement records of bead motility show successive runs including stalling (inset) and releasing events. (**D**) The peak force distributions and statistics (mean ± SEM) of stalling and releasing events in *pf18* cells. IFT particles exert 25–30 pN of peak forces with slightly less force produced in the anterograde direction.

The following figure supplements are available for figure 6:

**Figure supplement 1**. Three additional examples of simultaneous bead trapping and IFT27-GFP fluorescence tracking.

**Figure supplement 2**. The offset between the bead position and IFT trains.

**Figure supplement 3**. Additional examples for bead motility under fixed trap.

**Figure supplement 4**. The effect of antibody concentration on peak forces measurements.

# Materials and methods

## Strains and culture conditions

Vegetative *C. reinhardtii* cells were grown in Tris-acetate-phosphate (TAP) media in an illuminated plant growth chamber at 22°C. WT mt+ (cc125), *pf18* mt– (cc1297), and *pf18* mt+ (cc1036) strains were obtained from the *Chlamydomonas* Resource Center. The IFT20-GFP ΔIFT20 strain was provided by K Lechtreck and G Witman. The IFT27-GFP mt+ strain and *pf1 fla10-1*[ts] were provided by J Rosenbaum. IFT27-GFP *pf18* strain was generated through crosses (*Engel et al., 2009b*). *pf1 fla10-1*[ts] cells were incubated at 37°C for 15 min for heat inactivation of kinesin-2, and the movies were recorded within 30 min at 37°C, before the complete impairment of IFT and the start of flagellar

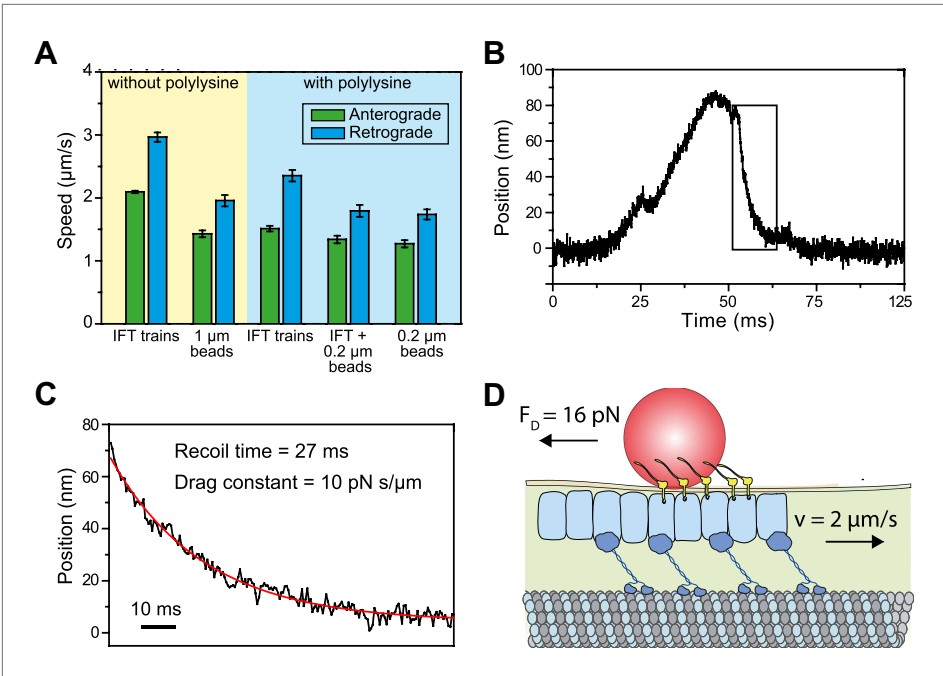

**Figure 7**. Viscous drag of the membrane slows down the motility of IFT trains that carry beads. (**A**) The average speed of FMG1-B antibody-coated beads and IFT trains. Coating of the coverslip surface with 0.7 mg/ml polylysine led to a ~20% reduction in the speed of IFT motility (two tailed *t*-test, p=5.2 × 10$^{-14}$ [anterograde] and 5.6 × 10$^{-6}$ [retrograde]). IFT trains that carry 0.2-μm beads move at 10–30% slower speeds than IFT trains that are not associated with the beads (p=1.1 × 10$^{-2}$ [anterograde] and 7.2 × 10$^{-5}$ [retrograde]). IFT trains that carry 0.2-μm beads move at similar speeds to beads (p=0.4 [anterograde] and 0.6 [retrograde]) (error bars represent SEM). (**B**) A typical example for stalling of bead motility by the optical trap. The bead recovers slowly to the trap center after a stall (rectangular box). (**C**) The recoiling of a trapped bead to the trap center after a stall was fitted to single exponential decay. The mean drag constant of the bead-FMG1-B-IFT complex is 8.0 ± 0.28 pN s/μm. (**D**) The IFT-bead complex experiences large drag forces as the IFT train moves several micrometers per second inside the flagellum (not to scale).

The following figure supplements are available for figure 7:

**Figure supplement 1**. Measurement of viscous drag on bead movement along the surface of the flagellar membrane.

resorption (*Kozminski et al., 1993*). In the *dhc1b-3*$^{ts}$ mt- mutant strain (cc4422) (*Engel et al., 2012*), dynein-1b is inhibited after 6 hr incubation at 37°C. Paralyzed flagella (*pf*) strains and paralyzing compounds (*Engel et al., 2011*) were utilized to impair swimming motility. Ciliobrevin D was dissolved in DMSO and then added to the *pf18* cell culture to inhibit dynein activity. IFT and gliding motility were recorded simultaneously 2 min after addition of 0–150 μM ciliobrevin D to the flow chamber.

## Single-molecule fluorescence imaging

The IFT and FSM imaging assays were performed with an objective-type TIRF microscope (*DeWitt et al., 2012*). The GFP signal was recorded by an electron multiplied charge-coupled device (EM-CCD) camera, with an effective pixel size of 106 nm. IFT27-GFP *pf18* and IFT20-GFP ΔIFT20 strains were used for tracking individual IFT trains (*Engel et al., 2009a*). The sample chamber was pre-treated with 0.7 mg/ml poly-L-lysine to adhere flagella on a glass surface. 15 μl cells in TAP media were applied to cover glass and inverted onto slides. Gliding motility assays were performed without polylysine treatment and cells were imaged under bright-field illumination. *ts* mutants were assayed at permissive (22°C) and restrictive (37°C) conditions by controlling the temperature with an objective heater (Bioptechs). Because paralyzed cells glide 30–40% faster than non-paralyzed cells (*Bloodgood, 1995*), gliding speeds of *ts* mutants of kinesin-2 (*pf1 fla10-1*$^{ts}$) and dynein-1b (*dhc1b-3*$^{ts}$) were compared to those of both *pf18* and WT cells at permissive (22°C) and restrictive (37°C) temperatures.

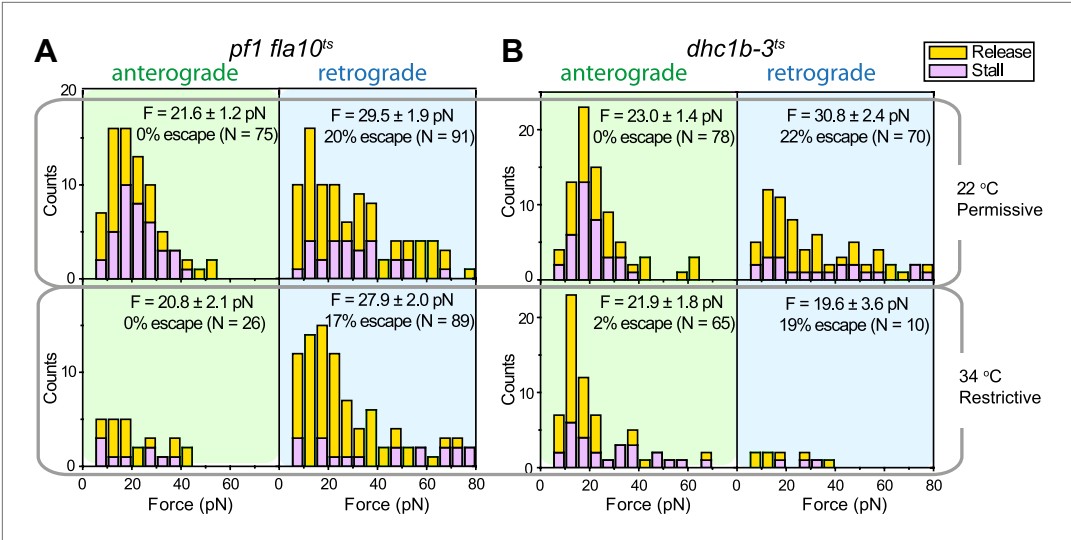

**Figure 8**. Force measurements on temperature-sensitive mutants. (**A**) Peak force histograms for IFT movement in *pf1 fla10$^{ts}$* cells at 22°C and 34°C. Stalling events are less common than the release of the bead. The frequency of anterograde IFT trains is reduced after switching the temperature to 34°C. Forces along the retrograde direction remain unaltered after the inhibition of kinesin-2 (mean ± SEM). (**B**) Peak force histograms for IFT in *dhc1b-3$^{ts}$*. The frequency and force production of retrograde IFT trains are reduced after switching the temperature to 34°C. Forces along the anterograde direction remain unaltered after the inhibition of dynein-1b (mean ± SEM).

## Bead-motility imaging

To visualize bead movement along the flagellar surface, 200 nm carboxyl-modified nile-red fluorescent beads (Invitrogen) were coated with anti-FMG1-B antibody (*Bloodgood et al., 1986*) by EDC-NHS cross linking. 1 ml *pf18* cell culture was spun down at 400*g* for 1 min and resuspended in 100 µl fivefold diluted TAP media. 1 µl bead stock solution (25 mg/ml) was then added to 20 µl of resuspended cell culture and the mixture was incubated in ice for 10 min, in order to slow down IFT movement for bead attachment. 1 µl of 20 mM $CaCl_2$ was then added, the cell culture was incubated at room temperature for 10 min for bead movement to recover. To immobilize cells on a glass surface, the cover glasses (18 × 18 mm) were pre-treated with 0.7 mg/ml poly-L-lysine (MW = 150–300 kDa; Sigma) for 5 min. The nile-red bead fluorescence was recorded with Andor iXon 128 × 128 EM-CCD at 400 µs per frame. To prevent deflagellation of cells under intense (100 mW) laser illumination, only the flagellar regions of immobilized cells were excited. Flagellar membrane adsorption of carboxylated 100 nm-beads was significantly reduced without antibody crosslinking (~50-fold), and only two bead motility events were observed in 500 cells.

## Multicolor-tracking assays

To simultaneously monitor the movement of IFT trains and FMG1-B proteins, IFT27-GFP *pf18* cells were incubated with anti-FMG1-B-coated dark-red beads. The GFP and dark-red bead fluorescence were simultaneously recorded at 200 ms per frame. The image was split into two fluorescent channels, which were registered to a sub-pixel accuracy (*DeWitt et al., 2012*) with respect to each other, prior to live-cell imaging. The crosstalk between the two fluorescent channels was below 0.1%.

Various concentrations of EGTA were added to IFT27-GFP *pf18* cells grown in TAP media (contains 0.34 mM of $Ca^{2+}$) and the movies were recorded within 15 min after EGTA addition. The concentration of free $Ca^{2+}$ in the assay buffer as a function of added EGTA was calculated from the Chelator program (http://maxchelator.stanford.edu). Only the cells with fully grown flagella (8–12 µm in length) were analyzed. Immotile IFT trains from the beginning to the end of the image acquisition were excluded from data analysis. The pausing frequency was calculated from dividing the total number of pausing events by the image acquisition time.

## Optical trapping assays

Force measurements on bead motility were carried out with a custom-built optical trap microscope (*Gennerich et al., 2007*), with single molecule fluorescence detection ability. A 1064 nm laser (Coherent, Compass) was mounted on an inverted microscope equipped with a 100× 1.49 NA oil immersion objective (Nikon). The trapping beam was steered by a computer-controlled acousto-optic deflector (AA Electronics) at 20 kHz. Trap stiffness was calibrated for each bead from the amplitude of its thermal diffusion. The beads were trapped by a 400 mW 1064 nm laser beam to achieve an average spring constant of ~0.4 pN/nm. The bead displacement was detected by a quadrant photodiode (QPD) and recorded at 2 kHz. Experiments were carried out with the trap position fixed. Stall forces were defined as the magnitude of the opposing force that reduces the mean velocity of the cargo to 0 for more than 100 ms. Returning events with stalling periods shorter than 100 ms at peak forces were scored as releases. Movements of beads beyond the linear range of the detector (±200 nm, ±80 pN) were scored as escapes. The trapping assays at restrictive conditions were performed at 34°C. Optical trap microscope was equipped with a 488 nm laser beam with near-TIRF excitation to track individual trains in IFT27-GFP cells. Bright-field illumination was turned on for ~1 s at every 1–2 min to monitor the bead position along the flagella and to synchronize the PSD signal and CCD image.

Carboxylated latex beads (0.92 µm diameter, Invitrogen) were coated with 0.05 mg/ml anti-carbohydrate mouse monoclonal antibody to FMG1-B for flagellar membrane attachment. Cover glasses were pre-treated with 0.7 mg/ml polylysine for 5 min. 10 µl *pf18* cell culture was flowed into a flow chamber and incubated for 30 s to allow cells to attach to the cover glass. 10 µl solution including 0.2× TAP, 1 mM CaCl$_2$, 0.3 mg/ml BSA, and 0.1 g/l beads was then flowed into the chamber to replace the buffer. Trapped beads were positioned over surface immobilized flagella (*Guilford and Bloodgood, 2013*). The fraction of moving beads was measured by resting beads on the flagellar surface for 1 min.

Viscous drag constant between the bead and the flagellar membrane was measured by oscillating a flagellar surface attached bead ±500 nm along the length of flagellum in a square wave pattern at 0.2 Hz. The connection between the bead and FMG-1B was verified by moving the trap away from the flagellum. If the bead displayed active movement powered by IFT trains during bead oscillation measurements, the bead was lifted for 0.5–1.0 µm to dissociate it away from IFT and then brought back to flagella.

## Data analysis

Speeds were measured from linear fits to the displacement traces observed in kymographs. In order to distinguish between IFT trains moving in anterograde and retrograde directions and IFT trains pausing over a period of time, we assigned different colors to individual IFT trains as a function of their velocity by Fourier space directional analysis (*Figure 2—figure supplement 2*). The image was then transformed back into real space to recover the kymographs. To perform IFT kymography analysis during gliding motility, we separately determined the position of IFT trains relative to the cell body, and the position of the cell body relative to the glass surface (*Figure 4*). Correlation between the movements of beads and IFT trains was demonstrated by rejecting the null hypothesis (IFT and bead movement was uncorrelated) using Kolmogorov-Smirnov statistics.

## Acknowledgements

We thank RD Vale and RA Bloodgood for evaluation of the manuscript, and M Porter, K Lechtreck, G Witman and J Rosenbaum for providing *C. reinhardtii* mutant strains and antibodies.

## Additional information

### Funding

| Funder | Grant reference number | Author |
| --- | --- | --- |
| National Institutes of Health | GM094522 | Ahmet Yildiz |
| National Institutes of Health | GM097017 | Wallace F Marshall |

| Funder | Grant reference number | Author |
| --- | --- | --- |
| National Science Foundation | MCB-1055017 | Ahmet Yildiz |
| Burroughs Welcome Foundation | | Ahmet Yildiz |
| Hellman Faculty Fund | | Ahmet Yildiz |

The funders had no role in study design, data collection and interpretation, or the decision to submit the work for publication.

## Author contributions

SMS, Conception and design, Acquisition of data, Analysis and interpretation of data, Drafting or revising the article; BDE, Analysis and interpretation of data, Drafting or revising the article, Contributed unpublished essential data or reagents; FK, AG, Acquisition of data, Drafting or revising the article; TB, Conception and design, Acquisition of data; WFM, Conception and design, Drafting or revising the article; AY, Conception and design, Analysis and interpretation of data, Drafting or revising the article

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
