## [Decision Letter]

Thank you for sending your work entitled “Intraflagellar Transport Drives Flagellar Surface Motility” for consideration at *eLife*. Your article has been favorably evaluated by a Senior editor and 3 reviewers, one of whom is a member of our Board of Reviewing Editors.

The following individuals responsible for the peer review of your submission want to reveal their identity: Suzanne Pfeffer (Reviewing editor), and William Hancock and Robert Bloodgood (peer reviewers).

The Reviewing editor and the other reviewers discussed their comments before we reached this decision, and the Reviewing editor has assembled the following comments to help you prepare a revised submission. All the comments can likely be addressed by clarification rather than any further experimental work.

1) Please clarify the viscosity measurement. This measurement is important because it implies that there are 16–24 pN viscous drag loads on the train-glycoprotein complexes, which is a significant insight. The observation is that when trapped beads attached to moving trains stall and then release, the bead is pulled back to the center of the trap relatively slowly (if it was simply a bead in solution, the bead drag forces are such that it would return within 1ms presumably). From the exponential fit and known trap stiffness they calculate the drag coefficient. One question is, what is the mode of failure? Is it motors detaching from the microtubules or could it be glycoprotein detaching from the train? (It seems that the simultaneous imaging says that the fluorescent train comes back with the bead, but we were not able to find this.) If the train is moving back, then the assumption is that the motors don't interact with the filament at all, but dynein has been shown to backstep and it's hard to rule out other motor-microtubule interactions, so the observed viscosity may not be just from the membrane. A possible experiment would be to step or oscillate a bead bound to transmembrane glycoprotein (but not train) and pull out viscosity that way.

2) The diagram of the cell body in Figure 3 was really helpful for orienting the transport directions. Can such a diagram be added to the other figures as well to orient the reader?

3) It is quite difficult to see the pausing in Figure 2. In the figure supplement there are some clearer examples, but the yellow highlighting in 2a makes it quite difficult to make out the pauses. The pausing is abundantly clear and quite convincing in Video 2. If the trains of interest were tracked and their position highlighted in a side-by-side Figure in 2a or possibly overlaid, that might help the reader see the paused trains. Figure 2 is difficult to understand. The odd coloring of the background is distracting, the red and blue arrows in the right panel are too small to see, and it's not described well. Perhaps thresholding to clean it up would help.

4) For Figure 2—figure supplement 2, specify this is magnitude of FFT (if this is correct). Figure 2–figure supplement 5 was very informative. Can it be moved to the main text. Also, the i) and ii) notations on the kymograph aren't referenced in the legend or the adjacent cartoon, where they apply.

5) “The data indicate that multiple motors attached to an IFT train may not be able to reach their maximum stall force”: this statement is not well supported by the data.

6) Figure 5 is confusing. The bead seems to move anterograde and then retrograde and then escape, but just before escape it seems to be settling to its original position. Is the position actually the QPD signal and not the real position in those cases around +/- 200 nm because that is the extent of the linear range of the QPD? Please clarify.

It is interesting that for wt the gliding speed is independent of temperature (22 vs 37) – shouldn’t motor speed vary with temperature?

7) Video 1: why does the level of red fluorescence appear to vary so much from one microsphere to another in the field of view?

8) The caption to Figure 5 gives the incorrect impression that the typical method of reversal of the direction of gliding motility of a *Chlamydomonas* involves the lifting of one flagellum off the substrate. While this can be sometimes observed, the vast majority of cases of reversals of direction are accomplished while both flagellar remain rigid and in contact with the substrate (albeit to different degrees).

---

## [Author Response]

*1) Please clarify the viscosity measurement. This measurement is important because it implies that there are 16–24 pN viscous drag loads on the train-glycoprotein complexes, which is a significant insight. The observation is that when trapped beads attached to moving trains stall and then release, the bead is pulled back to the center of the trap relatively slowly (if it was simply a bead in solution, the bead drag forces are such that it would return within 1ms presumably). From the exponential fit and known trap stiffness they calculate the drag coefficient. One question is, what is the mode of failure? Is it motors detaching from the microtubules or could it be glycoprotein detaching from the train? (It seems that the simultaneous imaging says that the fluorescent train comes back with the bead, but we were not able to find this.) If the train is moving back, then the assumption is that the motors don’t interact with the filament at all, but dynein has been shown to backstep and it’s hard to rule out other motor-microtubule interactions, so the observed viscosity may not be just from the membrane. A possible experiment would be to step or oscillate a bead bound to transmembrane glycoprotein (but not train) and pull out viscosity that way*.

If dynein steps backward during the recoiling of the bead, we would expect to observe backward steps and the velocity of the bead to be similar to that of dynein. However, the average velocity of the bead is 20 µm/s (an order of magnitude faster than that of dynein 1b) and there are no backward steps detectable, which rule out the possibility of dynein backtracking.

To rule out the possibility that our optical trapping measures glycoprotein rupture forces from IFT trains, we performed simultaneous optical trapping of the bead and fluorescence tracking of IFT trains. Figure 6 shows that IFT train underneath the bead (red dashed curve) recoils with the IFT train. Figure 6—figure supplement 1 also show that the bead makes multiple runs and recoils back to the trap center as an IFT train remains underneath the trapped bead, before the bead escapes the trap. The data indicate that we can measure the force production of the same IFT train multiple runs without rupturing the glycoprotein/bead linkage.

We performed the experiment the reviewers suggested. We oscillated a bead on a flagellar membrane surface in a square wave pattern and measured the recoiling time of the bead. The average viscous drag coefficient was 8.4 ± 4.2 pN.s/µm, similar to the drag that beads experience when they interact with IFT trains (8.0 ± 2.8 pN.s/μm). We added a short discussion about these results in the main text and showed the results in Figure 7—figure supplement 1.

*2) The diagram of the cell body in*
Figure 3
*was really helpful for orienting the transport directions. Can such a diagram be added to the other figures as well to orient the reader*?

We have added a diagram of a *Chlamydomonas* cell adhered to surface with both of its flagella to kymographs to other figures.

*3) It is quite difficult to see the pausing in*
Figure 2*. In the figure supplement there are some clearer examples, but the yellow highlighting in 2a makes it quite difficult to make out the pauses. The pausing is abundantly clear and quite convincing in*
Video 2*. If the trains of interest were tracked and their position highlighted in a side-by-side Figure in 2a or possibly overlaid, that might help the reader see the paused trains.*
Figure 2
*is difficult to understand. The odd coloring of the background is distracting, the red and blue arrows in the right panel are too small to see, and it’s not described well. Perhaps thresholding to clean it up would help*.

We thank the reviewers for their suggestions. In the revised manuscript, we included diagrams next to kymographs to highlight the paused IFT trains and gliding motility. Coloring in Figure 2 is removed, which makes red and blue arrows easier to see. Arrows are made bigger and extra description for these arrows are added in the figure legend.

*4) For*
Figure 2—figure supplement 2*, specify this is magnitude of FFT (if this is correct)*.

Yes. We specified the intensity as the magnitude of FFT in the figure legend.

*Figure 2–figure supplement 5 was very informative. Can it be moved to the main text. Also, the i) and ii) notations on the kymograph aren’t referenced in the legend or the adjacent cartoon, where they apply*.

We moved the figure to the main text and referred to the notations (shown in C) in D and the figure legend.

*5) “The data indicate that multiple motors attached to an IFT train may not be able to reach their maximum stall force”: this statement is not well supported by the data*.

This is one of the debated issues in multiple motors field. We clarified this statement by indicating the origin of discussion in the main text as follows: “Previous studies showed that multiple motors produce larger forces, move with higher velocities under load and carry cargos further than a single kinesin or dynein motor (37; 50). It remains controversial whether motor stall forces are additive at low motor copy numbers (50) or whether multiple motors tend to transport their cargo using only one load-bearing motor at a time (28). Our data show that the average forces exerted on IFT trains well exceed the force-generation capability of a single motor. [We did not observe a significant difference between the force values of stall and release events. The data indicate that multiple motors attached to an IFT train may not be able to reach their maximum stall force, defined as the stall force of a single motor multiplied by the number of bound motors (48).]”

*6)*
Figure 5
*is confusing. The bead seems to move anterograde and then retrograde and then escape, but just before escape it seems to be settling to its original position. Is the position actually the QPD signal and not the real position in those cases around +/- 200 nm because that is the extent of the linear range of the QPD? Please clarify*.

We chose this example in which the bead exceeded the linear range of the position sensitive detector (PSD) and escaped the trap. Additional examples with the beads stall/release within the linear range of PSD are provided in the Figure 6—figure supplement 1. In those cases, the movements of IFT trains are within 100 nm and can barely be observed in the fluorescence channel. The reviewers are right. The linear range of the PSD is roughly ±150 nm. As the bead moves further away from the trap center, the position calculated from the PSD signal decreases. To make this point clear, we are attaching a typical PSD response as a function of true distance between the bead and the trap center (Figure 9, below).Author response image 1PSD response curve as a function of bead-trap separation. The linear range of the PSD is ±150 nm. PSD signal decreases as the bead separation exceeds 200–300 nm.

We also show the rest of the bead movement trace, which shows the reduction in QPD signal, when the bead moves towards the extent of the linear range of the PSD (Figure 10, below).Author response image 2Simultaneous tracking of IFT27-GFP and bead motility. At t = 3 s, the microscope stage was moved to bring the flagellum underneath the trapped bead. An IFT train stalls at the trap position (t = 5–8 s) and stays adhered to the bead until the bead escapes the trap (t = 10 s). PSD signal shows a plateau and then decreases as the bead moves further away from the trap center, as shown in Figure 9.

*It is interesting that for wt the gliding speed is independent of temperature (22 vs 37) – shouldn’t motor speed vary with temperature*?

In vitro motility assays showed that the velocity of conventional kinesin motors increase significantly when the temperature is shifted from 22°C to 37°C in accordance with the Arrhenius rate law, but the force generation remain largely constant within a wide range of temperatures (e.g., Kawaguchi et al., BBRC, 2001). However, temperature dependence of *Chlamydomonas* dynein-1b and kinesin-2 velocity has not been explored in vitro. Most relevant studies were performed in in live *Chlamydomonas* cells, which showed that IFT speed does not vary between 22°C and 32°C (Lomini et al., JCB 2001). It remains unclear whether the speed of IFT motors is independent of temperature, or the gliding speed is limited by the net force production on paused IFT trains, which is temperature independent.

*7)*
Video 1*: why does the level of red fluorescence appear to vary so much from one microsphere to another in the field of view*?

First, small fraction of the 200 nm beads forms aggregates, and those aggregates tend to attach to the glass surface. These unusually bright beads were not included in the analysis of the data. Second, the sample is illuminated with TIRF, in which the illumination intensity decays exponentially by the z position of the object. The beads positioned on top of the flagella appear dimmer than the ones nonspecifically adhered to the glass surface.

*8) The caption to*
Figure 5
*gives the incorrect impression that the typical method of reversal of the direction of gliding motility of a Chlamydomonas involves the lifting of one flagellum off the substrate. While this can be sometimes observed, the vast majority of cases of reversals of direction are accomplished while both flagellar remain rigid and in contact with the substrate (albeit to different degrees)*.

We actually observed 60% of the time the cell lifted one flagellum to glide toward the other direction. We reckon that there can be several reasons why our observations are different from the previously published results. First, we use TIRF illumination, which is highly sensitive to the distance to the surface compared to brightfield imaging. Second, in certain cases, the cell only needs to lift a part of its flagellum to release the surface-attached IFT trains. We categorize these events as the cell lifts its flagellum to reverse gliding direction. We are not sure how these events were classified in previous studies. Third, different cell strains seem to have different tendency to lift their flagella. For example, *pf18* strain tends to lift one of its flagella during gliding less frequently. In our experiments, we used IFT27-GFP strain, which is different from the strains that are previously used for this experiment.